# Trans-differentiation of trophoblast stem cells: implications in placental biology

Madhurima Paul[1], Shreeta Chakraborty[1,2], Safirul Islam[1,3], Rupasri Ain[1]

**Trophoblast invasion is a hallmark of hemochorial placentation. Invasive trophoblast cells replace the endothelial cells of uterine spiral arteries. The mechanism by which the invasive trophoblast cells acquire this phenotype is unknown. Here, we demonstrate that, during differentiation, a small population of trophoblast stem (TS) cells trans-differentiate into a hybrid cell type expressing markers of both trophoblast (TC) and endothelial (EC) cells. In addition, a compendium of EC-specific genes was found to be associated with TS cell differentiation. Using functional annotation, these genes were categorized into angiogenesis, cell adhesion molecules, and apoptosis-related genes. HES1 repressed transcription of EC genes in TS cells. Interestingly, differentiated TCs secrete TRAIL, but its receptor DR4 is expressed only in ECs and not in TCs. TRAIL induced apoptosis in EC but not in TC. Co-culture of ECs with TC induced apoptosis in ECs via extrinsic apoptotic pathway. These results highlight that (a) TS cells possess the potential to trans-differentiate into "trophendothelial" phenotype, regulated by HES1 and (b) trophoblast differentiation-induced TRAIL secretion directs preferential demise of ECs located in their vicinity.**

## Introduction

Cell fate determination and differentiation accompany an exquisite molecular orchestra that is still an active area of research. Trophoblast cells, recognized as parenchymal cells of the placenta, execute most placental functions, indispensable for successful pregnancy. They differentiate from multipotent trophoblast stem (TS) cells during development. Despite being recognized as the developmental counterpart of embryonic stem (ES) cells in the context of placental development, many facets of regulation of trophoblast development remained elusive.

In rodents and primates, specialized populations of trophoblast cells of the placenta invade the uterine stroma and establish relationships with uterine blood vessels supplying the placenta (Pijnenborg et al, 1981; Cross et al, 2002; Georgiades et al, 2002;

Shukla & Soares, 2022; Ain et al, 2003). Two populations of invading trophoblast cells can be identified: (i) interstitial and (ii) endovascular. Interstitial trophoblast cells penetrate through the uterine stroma and are often situated in perivascular locations, whereas endovascular trophoblast cells enter uterine blood vessels, where they replace endothelial cells (Pijnenborg et al, 1981; Ain et al, 2003; James et al, 2022). It has been proposed that the "trophoblastic vascular colonization" is an effective mechanism for removing maternal vasomotor control and thus dramatically augmenting the delivery of maternal resources to the placenta (Pijnenborg et al, 1981). This hallmark developmental event in effect creates flaccid, low-resistance blood vessels, known as spiral artery remodeling (Nandy et al, 2020), and is fundamental for optimal delivery of nutrients to the fetus.

Invasion of trophoblast cells into placental arteries is associated with either displacing or co-existing with the endothelial cells. As is obvious from the foregoing that the maternal vascular space in rodents and primates is very unique in a way that trophoblast cells and not endothelial cells line the maternal side of the vasculature (Wooding & Flint, 1994; Ain et al, 2003). In order for these modified placental arteries to function normally, the invading trophoblast cells must acquire endothelial cell phenotype coincident with the event of invasion. In humans, there are reports of alteration of integrin expression in invading trophoblast cells that enable them to "fake" as endothelial cells (Damsky & Fisher, 1998; Kaufmann et al, 2003; Khankin et al, 2010). It has been proposed that molecular mechanisms underlying "trophoblast vasculogenesis" may be shared with endothelial cell development and that expression of VEGFR2 by trophoblast cells is an example of "trophoblast vasculogenesis" (Rai & Cross, 2014). Yet another such example is the endothelial cell protein NOSTRIN that plays important role in differentiation of TS cells to invasive trophoblast giant cells (TGCs) (Chakraborty & Ain, 2017, 2018).

It is evident that trophoblast vasculogenic mimicry comprises at least two subsequent processes: (a) acquisition of endothelial phenotype by trophoblast cells possibly through trans-differentiation and (b) selective endothelial cell death induced by trophoblast cells. It has been established that uterine spiral artery remodeling by human extra-villous trophoblast cells leads to endothelial cell apoptosis

---

[1]Division of Cell Biology and Physiology, CSIR-Indian Institute of Chemical Biology, Kolkata, India    [2]National Institutes of Health, Bethesda, MD, USA    [3]School of Biotechnology, Presidency University, Kolkata, India

Correspondence: rupasri@iicb.res.in

through Fas/FasL interaction (Ashton et al, 2005; Kuo et al, 2019). However, there are no reports on the entire endothelial gene repertoire that is expressed by trophoblast cells as they differentiate. The molecular regulators in trophoblast cells that drive this important developmental event and trophoblast-secreted factors that might induce selective endothelial cell death are not known.

Mouse TS cells derived by Tanaka et al (1998) provide an excellent paradigm for elucidating trophoblast differentiation and cell fate decisions ex vivo (Ullah et al, 2008; Saha et al, 2015; Latos & Hemberger, 2016; Chakraborty & Ain, 2018; Chrysanthou et al, 2018; Chakraborty et al, 2020; Saha & Ain, 2020; Basak et al, 2021; Basak & Ain, 2022). In this study, we demonstrate that a small population of mouse TS cells is capable of acquiring a composite genotype representing both trophoblast cells and endothelial cells during their differentiation. We have elucidated differentiation-induced changes in trophoblast cell transcriptome, related to endothelial cell function. We have established the role of the transcription factor HES1 in reprogramming trophoblast cells to acquire endothelial phenotype. Importantly, we showed that differentiated trophoblast cells secrete TNFSF10 (TRAIL), which can selectively induce apoptosis in endothelial cells located in its close vicinity. These findings have broad biological implications, as trophoblast vasculogenic mimicry is a prelude to spiral artery remodeling. Inefficient spiral artery remodeling leads to placental insufficiency and pregnancy-associated disorders.

# Results

## TS cell differentiation is associated with trans-differentiation of a population of TS cells into endothelial phenotype

To assess the potential of TS cells to trans-differentiate and acquire endothelial cell phenotype, expression of three most potent endothelial cell markers cadherin 5 (CDH5), platelet endothelial cell adhesion molecule 1 (PECAM1), and endoglin (ENG) was tested by immunofluorescence staining. Expression of the endothelial markers was observed exclusively in differentiated trophoblast cells (Fig 1A–C). To analyze the percentage of TS cell population that acquires endothelial cell characteristics during differentiation, flow cytometric analysis was used. Because trophoblast cells are epithelial in nature, pan cytokeratin (Ck) was used to detect all types of trophoblast cells. CDH5, PECAM1, and ENG were used as endothelial markers. TS cells showed almost negligible population (0.1–0.2%) expressing both the trophoblast and endothelial markers, whereas most of the population expressed only Ck. However, among the differentiated trophoblast cells, 11.6% were double positive for Ck and CDH5 (Fig 1D and G), 36.2% were double positive for PECAM1 and Ck (Fig 1E and H), and 35.4% were double positive for ENG and Ck (Fig 1F and I). In line with this, human trophoblast JEG3 cells were induced to express mesenchymal phenotype. It was observed that there was ~22% increase in CD144 (human cadherin 5) and HLAG (human trophoblast marker) double-positive cells, as compared with control un-induced JEG3 cells (Fig S1A and B). Similarly, there was ~13% increase in CD105 (human endoglin) and HLAG co-

expressing JEG3 cells as compared with controls (Fig S1C and D). Furthermore, the differentiated murine trophoblast cells grown in the presence of growth factors (VEGF$_{165}$ and bFGF), which potentiate endothelial cell function (Chakraborty & Ain, 2017; Khan et al, 2017), showed an increased population of cells co-expressing trophoblast and endothelial markers, whereas no appreciable change was seen in case of TS cells (Fig S2A and B). There was a 23.2% increase in Ck and CDH5 (Fig S2C and D) double-positive cells and a 53.2% increase in ENG and Ck double-positive cells (Fig S2E and F). These findings provided an indication of differentiation-associated acquisition of endothelial markers by a population of trophoblast cells that is enhanced under conditions supporting endothelial cell function.

## Differential expression of endothelial cell–specific functional transcriptome marks TS cell differentiation

To further our understanding on trophoblast vasculogenic mimicry, we analyzed endothelial cell functional transcriptome during TS cell differentiation using a real-time PCR–based mouse endothelial cell biology RT$^2$ profiler array. Scatter plot revealed expression patterns of 84 genes, which are known to affect endothelial cell function, in TS cells and differentiated trophoblast cells (Fig 2A). Overall, 53 transcripts met the recommended cut-off reads (Ct ≤ 30) in at least one of the two groups (Table S1). Of these, after validation in transcript and protein levels in multiple biological replicates, 13 genes showed significant differential expression upon induction of differentiation (Fig 2B and Tables 1 and 2). These endothelial cell function–associated genes that were found to be truly regulated upon TS cell differentiation in all biological replicates were functionally annotated into three different categories that included angiogenesis-related genes, cell adhesion molecules, and genes involved in apoptosis (Fig 2B). Validation of the array using qRT–PCR showed significant ($P < 0.01$ or $< 0.001$) up-regulation of two and down-regulation of three angiogenesis-related genes, upon TS cell differentiation (Fig 3A). The up-regulated genes included angiogenesis-promoting chemokine *Cx3cl1* and receptor tyrosine kinase *c-Kit*. However, the proteases *Mmp9*, *Plau* and angiogenic cytokine *Vegfα* receptor *Kdr* showed significant ($P < 0.01$) down-regulation (Fig 3A). Endothelial cell–specific adhesion molecules showing significant ($P < 0.01$) up-regulation in differentiated trophoblast cells included *Cdh5*, *Pecam1*, and integrin *Itgβ3*, whereas *Col18a1* showed significant ($P < 0.01$) down-regulation (Fig 3B). Interestingly, TS cell differentiation was also associated with differential regulation of apoptosis-related genes. There was a significant ($P < 0.01$) up-regulation of the death-inducing chemokine *Tnfsf10* (TRAIL) upon TS cell differentiation. Antiapoptotic gene *Bcl2* showed decreased ($P < 0.01$) expression (Fig 3C), whereas the proapoptotic adaptor protein *Cradd* and the executioner caspase *Casp3* also showed significantly ($P < 0.5$) decreased expression in differentiated trophoblast cells (Fig 3C).

To correlate transcript levels with protein expression, we further evaluated protein levels either by immunoblot assay or ELISA. Concordant with mRNA regulation, the angiogenesis-related protein KIT showed significant ($P < 0.01$) up-regulation, whereas proteases, such as MMP9 ($P < 0.01$), PLAU ($P < 0.01$), and the VEGF-$α$ receptor KDR, showed down-regulation ($P < 0.05$) (Fig 4A and B).

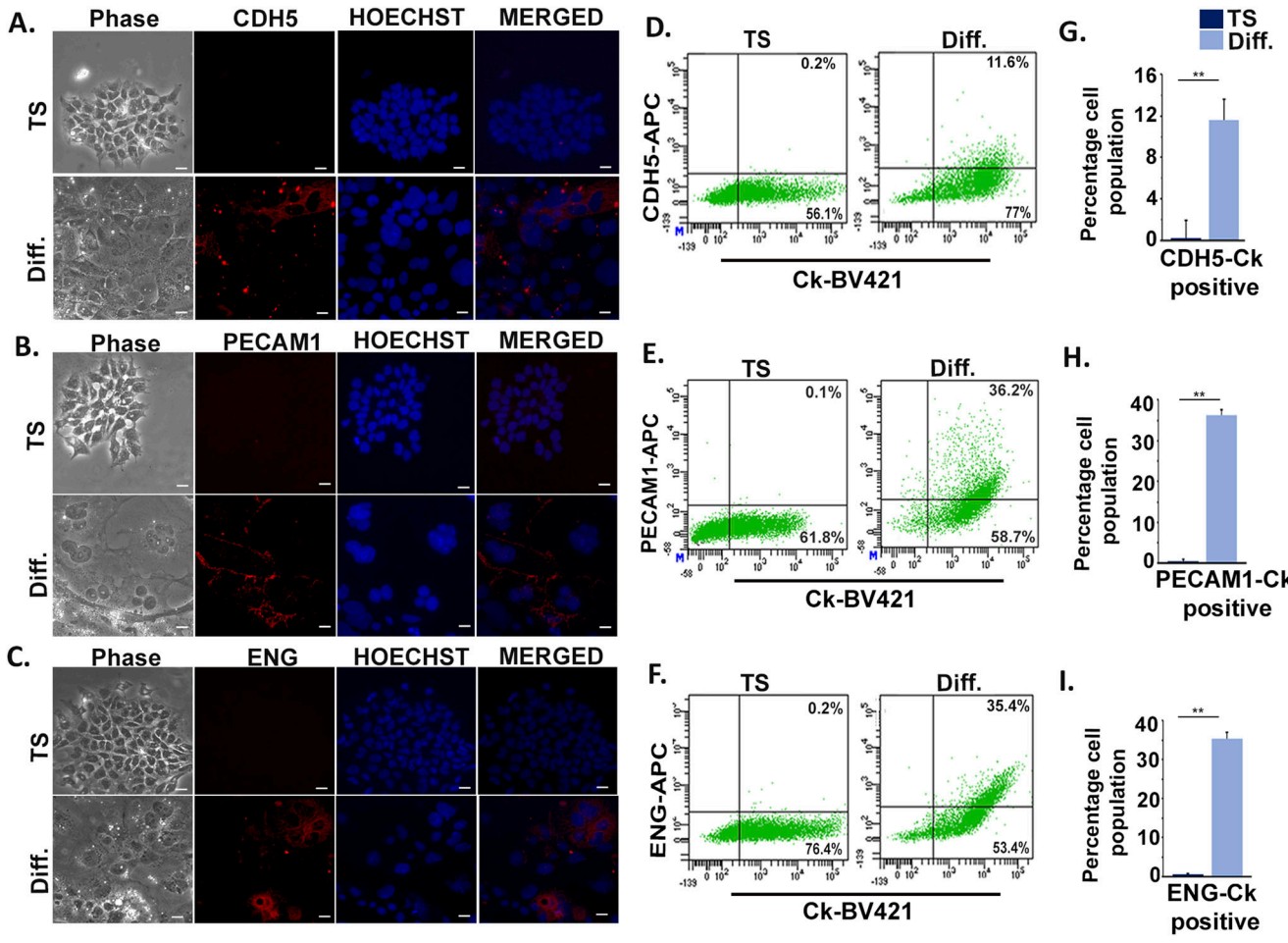

**Figure 1.   Induction of trophoblast stem (TS) cell differentiation is associated with trans-differentiation of a population of TS cells into endothelial phenotype.**
**(A, B, C)** Immunofluorescence staining of TS cells maintained for 6 d in stemness condition and day 6–differentiated cells (TC) using antibodies against endothelial markers, CDH5 (A), PECAM1 (B), and endoglin (C). Nuclei were counterstained with Hoechst. Scale bar represents 20 μm. Images were taken at a magnification of 200×.
**(D, E, F)** Flow cytometric analysis of TS and TC dually stained with trophoblast marker cytokeratin (Ck) and the endothelial specific markers, CDH5 (D), PECAM1 (E), and endoglin (F). **(G, H, I)** Quantification of the percentage of trophoblast cells showing dual positive staining for Ck and CDH5 (G), PECAM1 (H), and endoglin (I). Data are representative of three biological replicates, and error bars represent SEM. **$P < 0.01$.
Source data are available for this figure.

Potent endothelial cell markers and cell adhesion proteins CDH5, PECAM1 showed significant ($P < 0.01$) up-regulation along with integrin ITGβ3 ($P < 0.01$). Like its transcripts, COL18A1 protein levels decreased upon differentiation (Fig 4C and D). Differentiated trophoblast cells also showed decreased ($P < 0.01$) expression of the apoptosis-related proteins BCL2, CRADD, and caspase-3 (Fig 4E and F). Interestingly, protein levels of the secreted chemokine TNFSF10, a proapoptotic death-inducing ligand, also known as TRAIL, increased significantly ($P < 0.01$) upon TS cell differentiation (Fig 4G). Similarly, pro-angiogenic chemokine CX3CL1 showed significant ($P < 0.01$) up-regulation upon TS cell differentiation (Fig 4H). Endothelial cell–specific protein expression pattern corroborated with their mRNA expression. Thus, the array results and their validation revealed endothelial cell–specific functional markers' expression upon TS differentiation. These results highlight that TS cell differentiation might aid in culmination of trophoblast vasculogenic mimicry during development.

## Transcription factor HES1 potentiates acquisition of endothelial cell–specific markers during TS cell differentiation

Bioinformatics analysis of the transcription factor–binding sites on promoter regions of the endothelial cell–specific genes, such as *Cdh5*, *Pecam1*, and *endoglin*, revealed the presence of putative binding sequence (CACNAG) for the transcription factor HES1 (Takebayashi et al, 1994). HES1 has been reported to maintain stem cell state in several cancers (Liu et al, 2015) and also reported to be expressed in murine (Gasperowicz & Otto, 2008) and human placenta (Lacko et al, 2014). Interestingly, HES1 was found to be abundantly expressed in TS cells, and its protein levels decreased by ~86.5% in differentiated trophoblast cells. Concurrently, a significant ($P < 0.001$) increase in CDH5, PECAM1, and ENG protein levels was evident upon TS cell differentiation (Fig 5A and B). An inverse relationship in the expression patterns of HES1 and the endothelial markers CDH5 and PECAM1 was evident with progression of

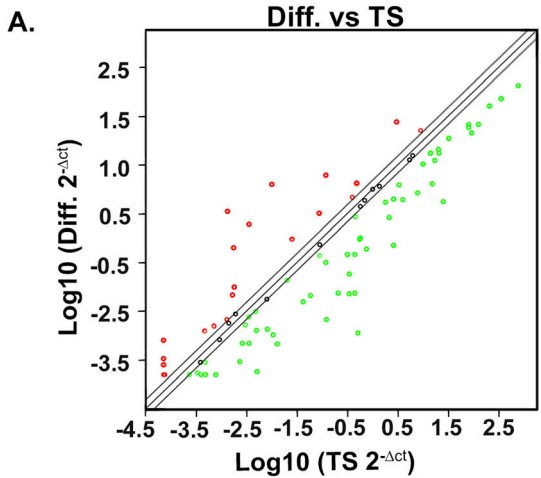

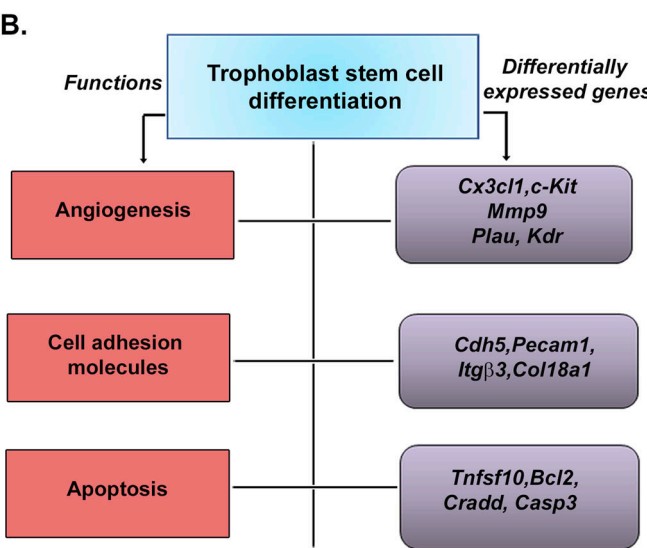

**Figure 2. Functional transcriptome analysis reveals acquisition of endothelial phenotype by trophoblast cells upon differentiation.**
**(A)** Scatter plot of the real-time PCR–based array of mouse endothelial cell–specific genes demonstrating differential expression patterns of 84 different genes from trophoblast stem cells (TS) and differentiated trophoblast cells (TC) on day 6 of differentiation. Normalization was done using the housekeeping gene that showed least change in an online software provided by SABiosciences. Up-regulated genes are marked by red dots, down-regulated by green, whereas those that remain unaltered are marked by black dots within regression lines.
**(B)** Schematic representation of the functional annotations of the 13 differentially regulated genes.

differentiation days (Fig S3). Furthermore, flow cytometric analysis revealed that a major population (10.8%) of differentiated trophoblast cells expressing the endothelial marker CDH5 were HES1 negative except a very small population of 2.8% cells that co-express CDH5 and HES1 upon differentiation (Fig S4). To further delineate probable role of HES1 in regulating endothelial cell–specific genes in TS cells, loss of function approach was used. RNA interference of *Hes1* using two Silencer Select siRNAs targeting *Hes1*-coding regions was performed in TS cells. The optimum concentration of the siRNAs used to knockdown *Hes1* was determined by a dose–response experiment (Fig S5). A final dose of

100 nM of total siRNAs (50 nM each) yielded a maximum of 60% *Hes1* mRNA down-regulation (Figs S5 and 5C) compared with the other dosages of 20 and 200 nM (Fig S1). Hence, this concentration was used for the further knockdown experiments. Hes1 knockdown in TS cells followed by 48 h of differentiation led to up-regulation of *Cdh5* and *endoglin* transcripts by 44% and 45%, respectively (Fig 5D and E). In line with this, precocious down-regulation of HES1 protein during TS cell differentiation led to significant ($P < 0.01$) up-regulation of CDH5 (1.3-fold) and ENG (2.24-fold) protein levels (Fig 5F and G). Furthermore, the influence of HES1 on acquisition of endothelial phenotype by trophoblast cells in the presence of endothelial function–promoting factors (50 ng/ml $VEGF_{165}$ and 10 ng/ml bFGF) was analyzed using flow cytometry. Interestingly, there was an increase in CDH5 and CK double-positive cells from 3.2% to 7.0% upon Hes1 RNA interference in the presence of $VEGF_{165}$ and bFGF (Fig S6A and B). Similarly, ENG and CK double-positive cells increased from 15.7% to 28.1% upon HES1 down-regulation in the presence of $VEGF_{165}$ and bFGF (Fig S6C and D). No appreciable change was observed in control scrambled siRNA-expressing cells for both the markers (Fig S6).

## HES1 binds to transcriptionally active promoter sites of endothelial cell marker genes in TS cells

Identification of putative HES1-binding sites on *Cdh5*, *Pecam1*, and *Eng* and the reciprocal expression pattern of HES1 with the endothelial markers CDH5, PECAM1, and ENG led to confirmatory experiment of HES1 binding to the promoter sites of these markers. Chromatin immunoprecipitation (ChIP) assay was performed in TS cells using HES1 antibody, followed by PCR analysis with different primer sets specific to four putative HES1-binding sequences (BS1–BS4) within 5 kb upstream to the transcription start site of each of the endothelial markers CDH5, PECAM1, and ENG, as has been represented schematically in Fig 6A, C, and E, respectively. ChIP-PCR assay confirmed HES1 binding to all the four binding sites in the promoter of CDH5 with intense bands for binding sites (BS) 1 and 2 (Fig 6B), although in case of PECAM1, promoter regions BS2 and BS3 showed intense bands compared with BS1, with no binding to the BS4 site (Fig 6D). However, HES1 binding was found to be restricted to the BS4 sequence of ENG, whereas very faint bands were seen for BS2 and BS3 (Fig 6F).

In addition, ChIP assay using RNA pol-II antibody confirmed that HES1-bound sites on the *Cdh5*, *Pecam1*, and *Eng* promoter regions were transcriptionally active (Fig 6B, D, and F). These results thus indicated that HES1 binding to the promoter regions represses transcription of the endothelial cell–specific genes in TS cells, and the repression is relieved upon decreased HES1 expression, with the induction of trophoblast differentiation.

## Differentiated trophoblast cells possess the potential to induce apoptotic death in endothelial cells through activation of extrinsic apoptotic pathway

Trophoblast vasculogenic mimicry is characterized by endovascular invasion of trophoblast cells into maternal arteries. These specialized populations of trophoblast cells either displace endothelial cells or are located underneath the vascular endothelium

**Table 1.  Differentiation-induced up-regulation of endothelial cell function genes in trophoblast cells.**

| Sl No. | Gene symbol | Accession no. | TS cell $C_t$ average | Diff. cell $C_t$ average | Fold change | Gene description |
|--------|-------------|---------------|------------------------|---------------------------|-------------|------------------|
| 1 | Tnfsf10 | NM_009425 | 30.76 | 23.84 | 261.99 | Tumor necrosis factor (ligand) superfamily, member 10 |
| 2 | Cdh5 | NM_009868 | 24.29 | 21.37 | 16.38 | Cadherin 5 |
| 3 | Pecam1 | NM_008816 | 24.87 | 22.49 | 7.89 | Platelet/endothelial cell adhesion molecule 1 |
| 4 | Kit | NM_021099 | 24.72 | 23.98 | 3.63 | Kit oncogene |
| 5 | Cx3Cl1 | NM_009142 | 25.02 | 23.88 | 2.75 | Chemokine (C-X3-C motif) ligand 1 |
| 6 | Itg$\beta$3 | NM_016780 | 22.25 | 21.92 | 2.72 | Integrin $\beta$ 3 |

(Rai & Cross, 2014). Transient co-existence of invaded trophoblast cells and endothelial cells in the modified maternal spiral arteries has been proposed earlier (Zhou et al, 1997; Adamson et al, 2002). Our results demonstrating secretion of TNFSF10 (TRAIL) by differentiating trophoblast cells led us to test whether TNFSF10 can induce apoptosis in trophoblast cells and/or endothelial cells. Induction of early apoptosis by increasing doses of TRAIL was assessed by annexin V–PI staining followed by flow cytometric analysis in MS1 endothelial cells and differentiated trophoblast cells. Endothelial cells underwent apoptosis in response to TRAIL exposure in a dose-dependent manner (Fig 7A and B), whereas no indication of apoptotic death induced by TRAIL was observed in differentiated trophoblast cells (Fig 7A and C).

To analyze the ability of differentiated trophoblast cells to induce apoptosis in endothelial cells, MS1 cells were co-cultured (on companion plates) with in vitro differentiated trophoblast cells (on inserts) for 48 and 72 h, so that two types of cells are physically separated but factors secreted by trophoblast cells can act on the MS1 cells. Apoptotic death after 48 h was observed in 19% of the co-cultured MS1 cells, whereas only 2.7% of control MS1 cells underwent apoptosis. Control MS1 cells were grown without the differentiated trophoblast cell–seeded inserts over them (Fig 7D). However, after 72 h of co-culture, apoptotic death increased to 30% compared with 0.1% control MS1 cells (Fig 7E). Furthermore, to verify whether this apoptotic death of the MS1 cells is TRAIL mediated, expression of the agonistic TRAIL receptor DR4 was tested using immunoblot analysis. Expressions of DR4 in control and co-cultured MS1 were comparable, whereas DR4 expression in differentiated

trophoblast cells was negligible (Fig 7F and G). Consequently, expression of potential apoptotic markers belonging to both the extrinsic and intrinsic apoptotic pathway in MS1 cells cultured in the presence or absence of differentiated trophoblast cells was analyzed using immunoblot assay. Significant ($P < 0.001$) increase in the active cleaved form of caspase-8 was observed in co-cultured MS1 cells as compared with controls. Similarly, an increase ($P < 0.5$) in cleaved caspase-3 was found in the co-cultured MS1 cells compared with controls (Fig 7H and I). Elevated cleaved caspase-8 and -3 indicate activation of the extrinsic apoptotic pathway in MS1 cells upon co-culturing with differentiated trophoblast cells. Interestingly, there was a reduction in the anti-apoptotic marker BCL2 belonging to the intrinsic apoptotic pathway member in the co-cultured MS1 compared with the control, although there was no significant change in the proapoptotic marker BAX (Fig 7J and K). These data indicate that differentiated trophoblast cells induce apoptotic death in a population of endothelial cells co-cultured with them through secretion of death-inducing ligand and subsequent activation of the apoptotic pathway.

Taken together, our findings indicate the potential of a population of TS cells to assume both endothelial and trophoblast phenotype (trophendothelial cells) driven by the transcription factor HES1. This phenomenon may aid in trophoblast vasculogenic mimicry during development. In addition, our investigation demonstrated that trophoblast-secreted TRAIL can induce apoptosis selectively in a population of endothelial cells but not in differentiated trophoblast cells. These data

**Table 2.  Differentiation-induced down-regulation of endothelial cell function genes in trophoblast cells.**

| Sl No. | Gene symbol | Accession no. | TS cell $C_t$ average | Diff. cell $C_t$ average | Fold change | Gene description |
|--------|-------------|---------------|------------------------|---------------------------|-------------|------------------|
| 1 | Col18a1 | NM_009929 | 22.75 | 29.44 | −47.58 | Collagen, type XVIII, alpha 1 |
| 2 | Mmp9 | NM_013599 | 16.54 | 23.2 | −46.53 | Matrix metallopeptidase 9 |
| 3 | Kdr | NM_010612 | 19.82 | 26.17 | −37.65 | Kinase insert domain protein receptor |
| 4 | Bcl2 | NM_009741 | 21.62 | 26.42 | −12.86 | B-cell leukemia/lymphoma 2 |
| 5 | Cradd | NM_009950 | 22.4 | 26.78 | −9.6 | CASP2 and RIPK1 domain containing adaptor with death domain |
| 6 | Plau | NM_008873 | 25.78 | 29.98 | −8.51 | Plasminogen activator, urokinase |
| 7 | Casp3 | NM_009810 | 22.08 | 25.75 | −5.86 | Caspase-3 |

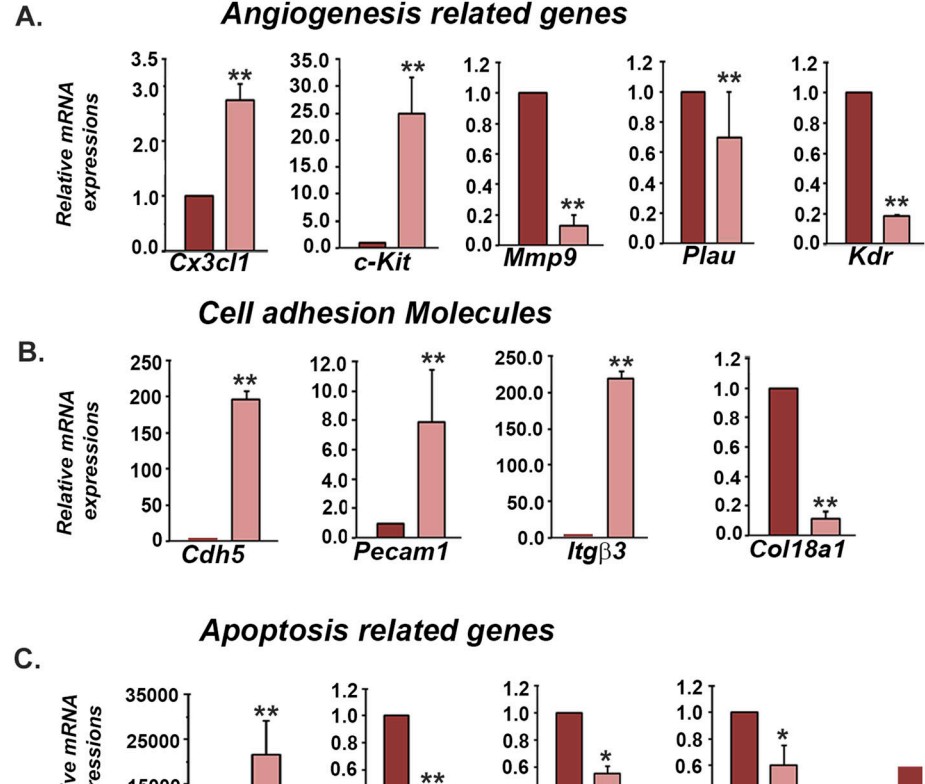

Figure 3.   Real-time PCR analysis of endothelial cell–specific transcripts that are regulated in trophoblast during differentiation.
**(A, B, C)** Quantitative real-time PCR (qRT) analysis of the differentially expressed endothelial cell–specific genes in trophoblast stem cells (TS) and differentiated trophoblast cells (TC). qRT–PCR results are grouped based on functional annotations. **(A)** Angiogenesis-related genes: *Cx3cl1,c-kit, Kdr, Plau, Mmp.* **(B)** Cell adhesion molecules: *Cdh5, Pecam1, Itgβ3, Col18a1.* **(C)** Apoptosis-related genes: *Tnfsf10, Bcl2, Cradd, Casp3.* Data are representative of three biological replicates, and error bars represent SEM. *P < 0.5, **P < 0.01.
Source data are available for this figure.

elucidate the mechanism of endovascular trophoblast invasion, which is an essential prelude to trophoblast vasculogenic mimicry.

# Discussion

Acquisition of endothelial character by the trophoblast cells during placental development is considered as a remarkable process of spiral artery remodeling, which is associated with erosion/de-differentiation of the smooth muscle cells lining the maternal arteries to regulate their vascular tone (Whitley & Cartwright, 2010; Rai & Cross, 2014; Nandy et al, 2020). These trophoblast cells are specialized subtypes of TGCs and are classified based on their developmental origin and function, and they have the potential to undergo morphogenesis to form vascular tubes (Simmons & Cross, 2005; Simmons et al, 2007; Knöfler et al, 2019). However, the molecular mechanism by which TGCs replace the endothelial cells lining the uterine arteries is yet to be elucidated in detail.

Our flow cytometric study has successfully established that a population of murine trophoblast cells, upon differentiation, express markers characteristic of both endothelial cells and trophoblast cells. Similar phenomenon was observed to be existing in human trophoblast cell line JEG3 that possesses properties of both placental extra-villous trophoblasts and trophectoderm (TE) stem cells (Dietrich et al, 2021). Detection of cell population co-expressing endothelial and trophoblast markers in un-induced JEG3 cells is indicative of the intrinsic property of invasive trophoblast cells (arising upon differentiation of TE stem cells in early placental events). Induction of mesenchymal phenotype potentiates their capacity to acquire both trophoblast and endothelial cell phenotype. It might be alluring to assume that a population of TS cells undergoes trans-differentiation into endothelial cells, but these TS cells that start expressing endothelial cell markers do not express the endothelial cell lineage–determining transcription factor ERG (data not shown). In addition, co-expression of genetic markers for both trophoblast cells and endothelial cells by these differentiated trophoblast cell populations prompted us to term these unique populations of trophoblast cells as "trophendothelial cells." Acquisition of "trophendothelial" phenotype was enhanced further in the presence of endothelial cell function–promoting factors, exclusively in differentiating trophoblast cells but not in TS cells. It is evident from our data that TS cells do not express the genetic markers unique to endothelial cells; therefore, our study clearly established that differentiation imparts the trophendothelial phenotype on a population of trophoblast cells.

There are many reports of expression of endothelial cell markers by trophoblast cells (Damsky et al, 1992; Damsky & Fisher, 1998; Pavličev et al, 2017). But our data on RT$^2$ profiler array followed by its validation at both transcript and protein levels established the

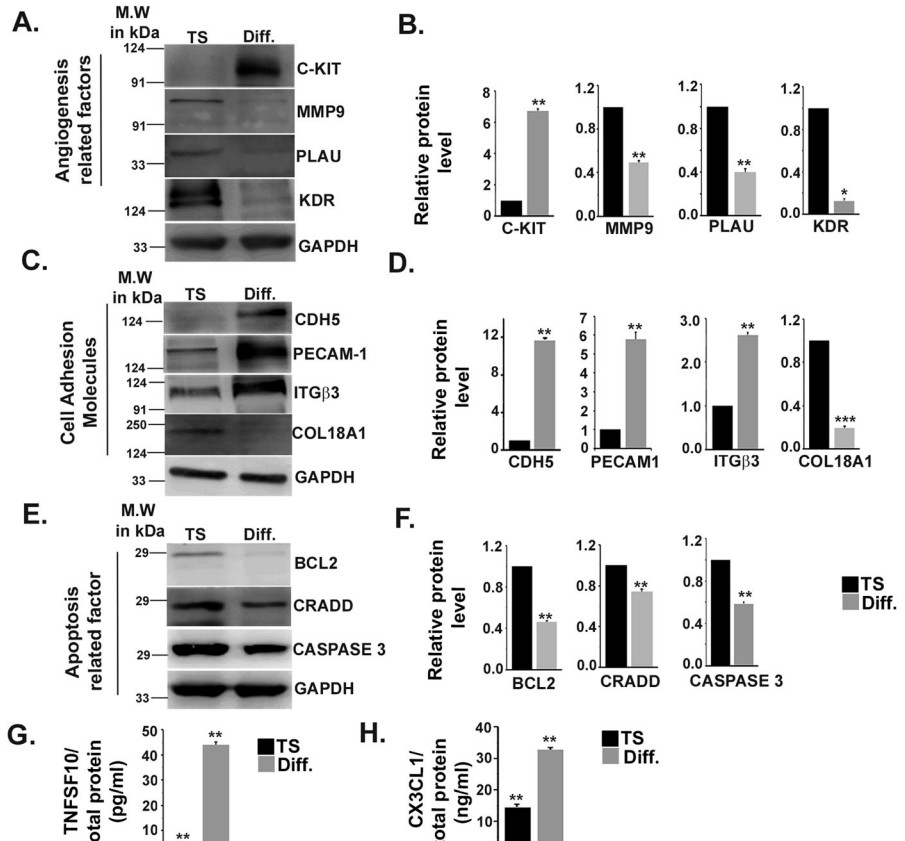

**Figure 4. Trophoblast differentiation marks differential expression of endothelial cell signature proteins.**
**(A, C, E)** Western blot analysis of proteins isolated from TS and TC showing differential expression of (A) angiogenesis-related markers: C-KIT, KDR, PLAU, MMP9, (C) cell adhesion molecules: CDH5, PECAM1, ITGβ3, COL18A1; (E) apoptotic protein, BCL2, CRADD, and caspase-3. **(A, B, C, D, E, F)** Densitometric analysis of blots from (A, C, E), respectively, using NIH ImageJ software. Normalization was done with GAPDH using three biological replicates. **(G, H)** ELISA analysis of secreted cytokines TNSF10 and CX3CL1, respectively. Cytokines were normalized to total cellular proteins. Error bars represent SEM. *P < 0.5, **P < 0.01. Source data are available for this figure.

compendium of endothelial cell functional genes that are expressed in trophoblast cells upon differentiation. These include members of angiogenesis-related factors, cell adhesion molecules, and factors related to apoptosis. Endothelial cell proliferation, cytoskeletal reorganization, migration, and formation of vascular lumen are promoted by angiogenic signals (Munoz-Chapuli et al, 2004), which includes release of chemokines, pro-angiogenic ligands and decreased production of anti-angiogenic factors. Our data demonstrated up-regulation of many such pro-angiogenic factors and their receptors like Cx3cl1 and c-Kit, along with down-regulation of proteases such as Plau and Mmp9, in differentiated trophoblast cells. These data indicate balanced endothelial functional potential of differentiated trophoblast cells.

Expression of endothelial cell–specific adhesion molecules upon trophoblast differentiation has remarkable functional connotation. Cell adhesion molecules form an important group of endothelial markers involved in homo- and/or heterophilic interaction with other cells or with the extracellular matrix (Goncharov et al, 2017). Differentiated trophoblast cells showed increased expression of such cell adhesion molecules, implying their acquisition of endothelial cell properties. Expression of CDH5 by differentiated trophoblast cells indicates potential of differentiated trophoblast cells to form tight junctions as they line up the maternal arteries. CDH5 is known to be a part of tight junction formation in endothelial cell linings (Sauteur et al, 2014). This finding is in line with previous reports of expression of CDH5 by invasive trophoblast cells

(Dubernard et al, 2005; Sung et al, 2022). Trophoblast-specific knockout of CDH5 in mice has been shown to cause inadequate spiral artery remodeling, leading to intra-uterine growth restriction and death of the fetus (Sung et al, 2022). Taken together, all these data highlight the functional importance of the trophendothelial cells during development.

Apoptosis is one of the primary processes involved in tissue remodeling, an important event during placental development (Naicker et al, 2013), and in endothelial cells, it plays a prominent role in blood vessel development, homeostasis, and remodeling (Affara et al, 2007). Secretion of death-inducing ligand TNFSF10 or TRAIL by the differentiated trophoblast cells (this report) raised the hypothesis that trophoblast cells are protected from TRAIL-induced apoptosis, whereas TRAIL might selectively induce apoptosis in endothelial cells located in vicinity of trophoblast cells. TRAIL is known to signal through the agonistic death receptor DR4 (Wang & El-Deiry, 2003). Our data on expression of DR4 by endothelial cells but not by differentiated trophoblast cells affirm the hypothesis of selective apoptosis. Although our data showed decreased expression of the antiapoptotic factor Bcl2, there was a simultaneous reduction in expression of the executioner caspase caspase-3 and also the FADD-like proapoptotic adaptor CRADD known to activate FAS/TNFR apoptotic pathway (Ahmad et al, 1997), indicating apoptotic death less is likely to occur in the differentiated trophoblast cells.

Hes1 is a known transcriptional repressor that plays pleiotropic roles in various aspects of cellular regulation including stem cell

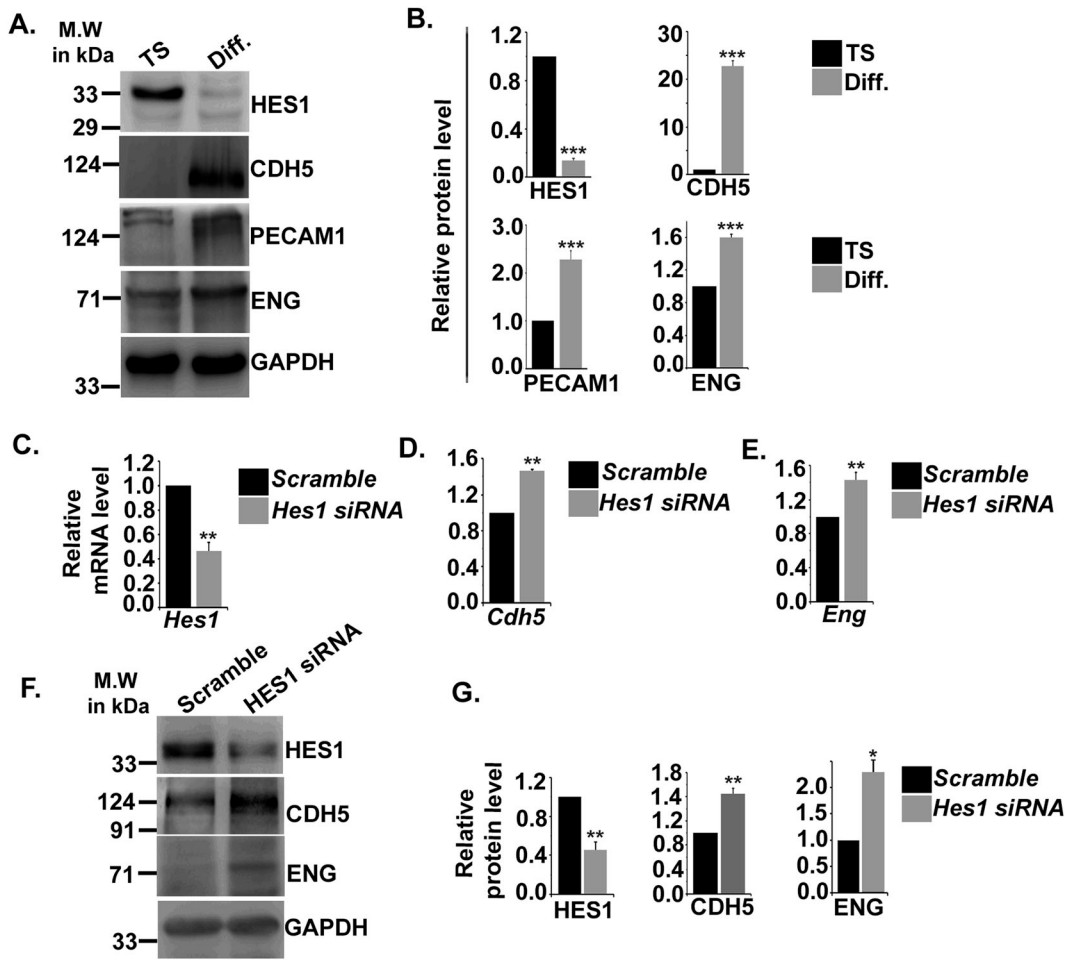

**Figure 5. Acquisition of endothelial markers upon trophoblast differentiation is associated with down-regulation in Hes1.**
**(A)** Western blot analysis for HES1 and endothelial cell–specific proteins CDH5, PECAM1, and ENG using cell lysates from TS and TC. **(A, B)** Densitometric analysis of the proteins from blots in (A) using NIH ImageJ software after normalization with GAPDH. **(C)** Quantitative real-time PCR of *Hes1* using RNA from TS cells transfected with either scrambled or *Hes1* siRNA. **(D, E)** Quantitative real-time PCR of *Cdh5* (D) and *endoglin* (E) using RNA from TS cells transfected with either scrambled or *Hes1* siRNA. **(F)** Western blot analysis of HES1, CDH5, PECAM1, and ENG using cell lysates from TS cells transfected with either 100 nM scrambled or *Hes1* siRNA followed by induction of differentiation till day 2. **(F, G)** Densitometric analysis of the proteins from (F) using NIH ImageJ software after normalization with GAPDH. Data are representative of three independent biological replicates. Error bars represent SEM. *P < 0.5, **P < 0.01.
Source data are available for this figure.

maintenance in several cancer cells (Liu et al, 2015; Cenciarelli et al, 2017). Hes1 is reported to be expressed in both human and mouse placenta (Gasperowicz & Otto, 2008; Lacko et al, 2014). Our data on reciprocal expression of Hes1 and the potent endothelial markers CDH5, PECAM1, and ENG during TS cell differentiation were an indication that Hes1 might repress transcription of these genes in TS cells. Interestingly, our flow cytometric data confirmed that CDH5-expressing differentiated trophoblast cells do not express Hes1. Data on precocious down-regulation of Hes1 during TS cell differentiation re-affirmed Hes1-mediated repression of these genes. In addition, the presence of endothelial cell function–promoting growth factors further increases "trophendothelial" cell population when the Hes1-mediated repression is relieved. Binding of HES1 to the transcriptionally active promoters of these endothelial genes in TS cells further indicates Hes1-mediated transcriptional repression of these genes in TS cells. Interestingly, bioinformatics analysis also

showed HES1-binding sites in the promoter regions of the death-inducing ligand TNFSF10. Our data on Hes1, inversely related expression of Hes1, and TNFSF10 during trophoblast differentiation indicate that Hes1 might repress TNFSF10 expression in TS cells as well. However, further investigation is required to establish regulation of Hes1-mediated TNFSF10 release by the differentiated trophoblast cells.

Trophoblast–endothelium interaction/cross-talk is said to be involved in the transformation of maternal spiral arteries, leading to their remodeling (Enders & Welsh, 1993; Adamson et al, 2002). Induction of endothelial cell apoptosis by extra-villous trophoblast cells in mixed culture has been reported (Ashton et al, 2005). This induction of apoptosis is proposed to be a consequence of the trophoblast–endothelium cross-talks in the maternal spiral arteries. Our data on selective apoptosis induction in endothelial cells by trophoblast-secreted TRAIL via extrinsic pathway expand

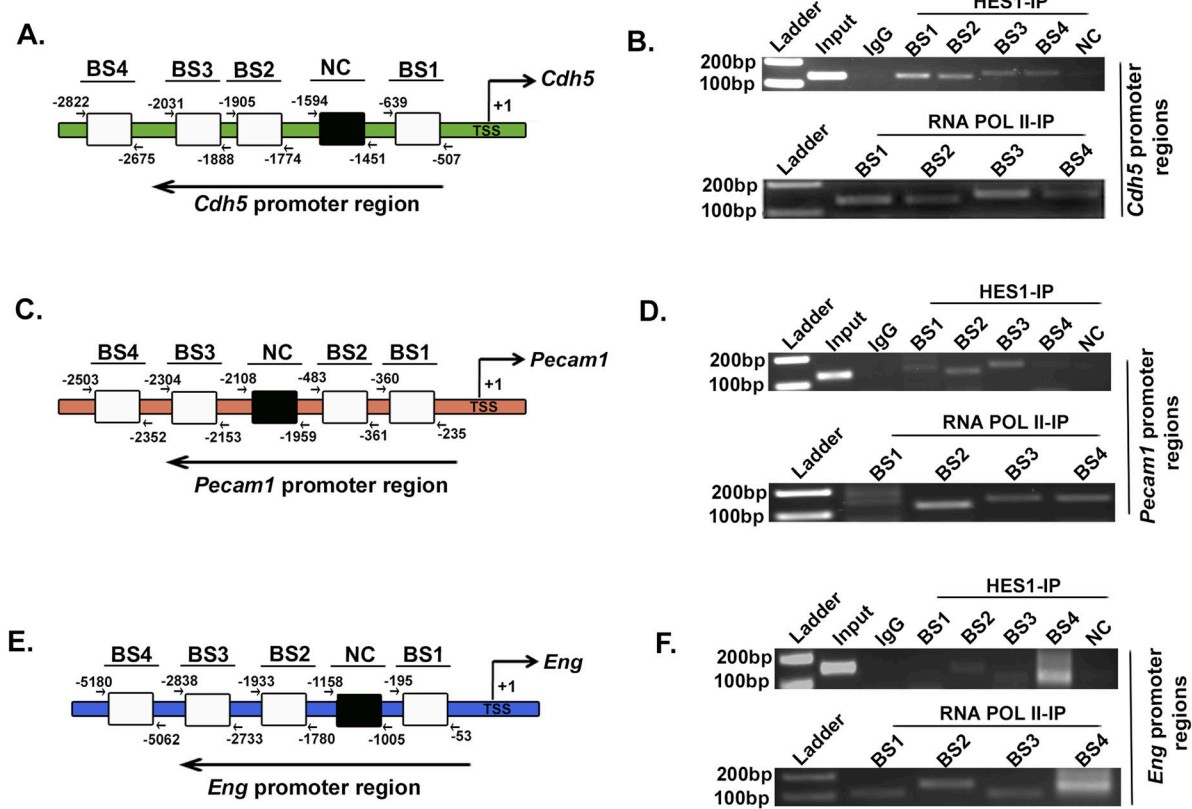

**Figure 6. Hes1 binds to transcriptionally active promoter sites of endothelial marker genes in trophoblast stem cells.**
**(A, C, E)** Schematic representation of the putative HES1-binding sites on promoter/enhancer region about 5 kb upstream to the transcription start site of the endothelial marker genes *Cdh5* (A), *Pecam1* (C), and *Eng* (E). **(B, D, F)** Chromatin immunoprecipitation assay with either HES1 or RNA polymerase II antibody using genomic DNA from TS cells followed by PCR analysis. BS, binding site.
Source data are available for this figure.

our knowledge on trophoblast–endothelium cross-talk. Findings from this study highlight that trophoblast vasculogenic mimicry involves acquisition of trophendothelial phenotype during TS cell differentiation, associated with trophoblast-induced selective apoptotic death of endothelial cells located in the vicinity of trophoblast cells.

# Materials and Methods

### Cell culture and differentiation

Mouse TS cells were gifted by Prof. Janet Rossant, Hospital for Sick Children, Toronto, Canada. TS cells were cultured in 30% TS complete media (RPMI-1640 [Sigma-Aldrich]) supplemented with 20% FBS (Invitrogen), 1% Penicillin-Streptomycin, 1% GlutaMAX (Gibco), 1 mM sodium pyruvate, 100 µM $\beta$ mercaptoethanol (Sigma-Aldrich)] and 70% MEF-conditioned medium supplemented with 25 ng/ml FGF4 (R&D Systems) and 1ug/ml heparin (Sigma-Aldrich) at 37°C in a humidified incubator with 5% $CO_2$. MEF-conditioned media were collected from mitomycin C–treated primary MEF cells isolated from E13.5 embryos of C57BL6 mice. Differentiation of TS cells was induced by withdrawal of mitogens, that is, FGF4, heparin, and

conditioned medium (Tanaka, 2006) and continued till 6 d post-induction unless otherwise mentioned.

To test the influence of endothelial cell function–promoting factors during differentiation, TS cells were cultured at a density of $10^4$/ml in either stemness or differentiated media, supplemented with 50 ng/ml $VEGF_{165}$ (293-VE/CF; R&D Systems) and 10 ng/ml bFGF (Sigma-Aldrich) and grown for 6 d with media change at an interval of 48 h.

In case of growing cells under conditions promoting endothelial character, trophoblast cells at a density were cultured at a density of $4 \times 10^4$/4 ml in either stemness or differentiated media, supplemented with endothelial growth factors, at a final concentration of 50 ng/ml $VEGF_{165}$ (293-VE/CF; R&D Systems) and 10 ng/ml bFGF (Sigma-Aldrich) and grown for 6 d with media change at an interval of 48 h.

Human trophoblast cell line JEG3 was obtained from American Type Culture Collection and grown in Eagle's Minimal Essential Medium (Sigma-Aldrich) supplemented with 10% fetal bovine serum (Invitrogen), 1% Penicillin-Streptomycin (Gibco), 1% GlutaMAX (Gibco), and 1 mM sodium pyruvate (Sigma-Aldrich). Mesenchymal phenotype was induced in JEG3 cells by growing them in the presence of 1× growth supplements (CCM017; R&D Systems) for 5 d, with media change at an interval of 48 h.

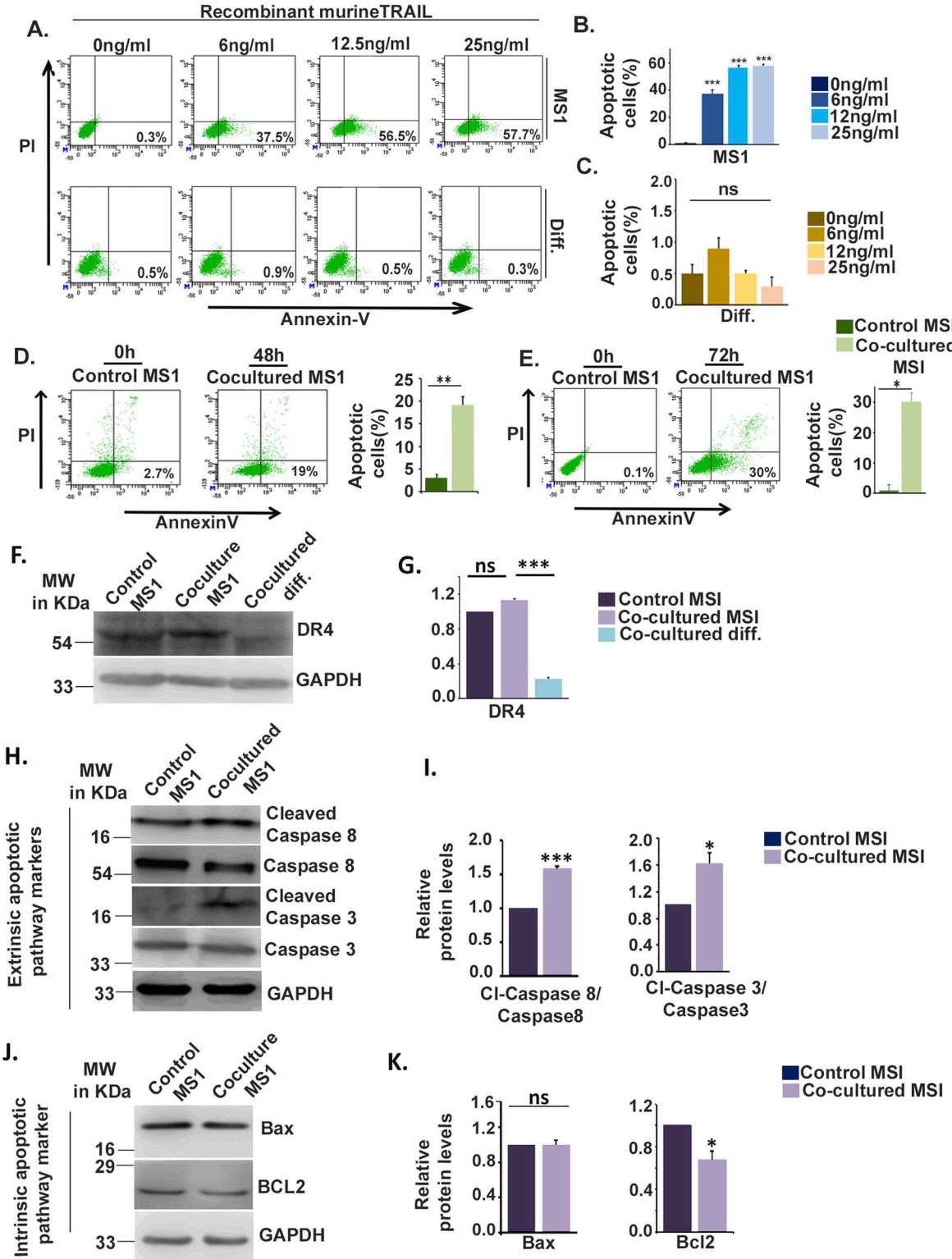

**Figure 7. Differentiated trophoblast cell induces apoptotic death of endothelial cells through activation of extrinsic apoptotic pathway.**
**(A)** Apoptotic death induced by increasing dose of TRAIL in endothelial cells (MS1) and differentiated trophoblast cells (TC) assessed using annexin V–PI–based flow cytometric analysis. Corresponding untreated cells were kept as controls. **(A, B, C)** Quantitative analysis of percentage of from (A) in MS1 cells and (C) TC has been shown in bar graphs. Data are representative of three independent biological replicates. Error bar represents SEM. ***$P < 0.001$; ns, nonsignificant. **(D, E)** Annexin V–PI–based flow cytometric analysis to assess apoptotic death of endothelial cells (MS1) co-cultured either in the absence or presence of differentiated trophoblast cells for 48 h (D) or 72 h (E). Quantitative analysis of percentage of cells undergoing apoptotic death has been shown in adjacent bar graphs. **(F)** Western blot analysis of the TNFSF10 (TRAIL)

Mouse endothelial cell line MS1 was obtained from American Type Culture Collection. Endothelial cells were grown in Dulbecco's Modified Eagle's Medium, high glucose (Sigma-Aldrich) supplemented with 5% fetal bovine serum (Invitrogen) in the presence of 1% Penicillin-Streptomycin (Gibco).

The Indian Institute of Chemical Biology Animal Ethics and Care Committee approved all procedures for handling and experimentation with rodents as per guidelines set forward by the Committee for the Purpose of Control and Supervision of Experiments on Animals (CPCSEA), Govt. of India (http://cpcsea.nic.in).

## RNA interference

For *Hes1* knockdown, TS cells were plated in a 35-mm dish 24 h before transfection. Cells at 60–70% confluence were transfected with either scramble (control) or equal concentrations of two pre-validated Silencer Select *Hes1*-siRNAs targeting the coding region of murine HES1 (assay ID: s67462 and s67463; Ambion) at a final concentration of 20, 100, and 200 nM using Lipofectamine RNAiMAX (Invitrogen) as per manufacturer's instructions. Differentiation was induced 6 h post-transfection and maintained for 48 h before RNA and protein isolation. Down-regulation was assessed at transcript level using quantitative real-time PCR and at protein level by WB analysis. Furthermore, knockdown experiments were performed using the concentration showing maximum down-regulation, 100 nM. For knockdown experiments, performed under conditions promoting endothelial cell function, differentiation was induced in TS cells 6 h post-transfection, and cells were maintained in differentiation media for 48 h in the presence of endothelial growth factors at a final concentration of 50 ng/ml $VEGF_{165}$ (293-VE/CF; R&D Systems) and 10 ng/ml bFGF (Sigma-Aldrich) before flow cytometric analysis of the trophoblast and endothelial markers.

## Immunofluorescence

TS and differentiated trophoblast cells were grown on glass coverslips. Cells were stained either in stemness (6 d after seeding) or differentiated state (6 d after induction of differentiation). Cells were fixed with 4% paraformaldehyde for 15 min at room temperature. Cells were then washed in 1× PBS (Gibco) and blocked with 5% serum (from 2° antibody host) in 1× PBS containing 0.3% Triton-X for 1 h at room temperature. After blocking, incubation with primary antibodies at recommended dilutions was done in antibody buffer 1× PBS containing 1% BSA and 0.3% Triton-X for 1.5 h at room temperature. Cells were then washed in 1× PBS three times and incubated with TRITC-conjugated secondary antibodies for 1.5 h at room temperature in the dark. Cells were then washed with 1× PBS three times and counterstained with Hoechst (2 µg/ml) in 1× PBS for 20 min in the dark. Cells were washed with 1× PBS five to six times

and mounted onto glass slides using fluoroshield mounting medium (Sigma-Aldrich). Stained cells were imaged using Leica DMi8 epifluorescence microscope at a magnification of 200×.

## Flow cytometric analysis of endothelial and trophoblast markers

TS cells cultured for 24 h in stem cell condition and day 6–differentiated trophoblast cells were harvested by trypsinization at a final concentration of $10^6$ cells/250 µl. For the experiments carried out under conditions promoting endothelial function, cells were cultured either in stemness (6 d after seeding) or differentiated state (6 d post-induction of differentiation) in the presence or absence of the endothelial growth factors and harvested as described above. Single-cell suspension was blocked using 5% serum (from 2° antibody host) in 1× permeabilization buffer (cat no. 00-8333-56; Invitrogen) at 4°C for 30 min. Cells were then washed once in 1× permeabilization buffer by centrifuging at 250*g* for 5 min. Subsequently, the cells were incubated with primary antibody in 1× permeabilization buffer at recommended dilutions for 2 h at 4°C. This was followed by washing the cells twice using 1× permeabilization buffer. Cells were then incubated with respective secondary antibodies in 1× permeabilization buffer at recommended dilutions for 1 h at 4°C. Post-incubation cells were washed twice using 1× permeabilization buffer and resuspended in 1× PBS (Gibco) followed by analysis in a flow cytometer (LSR Fortessa; BD) using appropriate filters for respective fluorochromes. For unstained cells, primary antibody incubation was substituted by 1× permeabilization buffer only, and rest of the steps remained identical to the processing of the stained samples.

## Reverse transcription and qPCR assay

Total RNA was isolated from cells using TRIZOL reagent (Invitrogen) as per manufacturer's protocol. Extracted RNA was reverse transcribed using the M-MLV Reverse Transcription kit (Invitrogen). Quantitative real-time PCR reaction was set in a 7500 real-time PCR system (Applied Biosystems) with a 10-fold dilution of cDNA and Power SYBR Green PCR Master Mix (Applied Biosystems) under standard PCR conditions as described previously (Chakraborty & Ain, 2017). Expression of endogenous *Rpl7* was used for normalization of genes of interest. Relative expression of RNA was calculated using $2^{-\Delta\Delta Ct}$ method. Primers used have been listed in Table S2.

## Quantitative RT² profiler PCR array

A large-scale quantitative real-time PCR array was performed using a mouse endothelial cell biology RT² Profiler PCR array (catalogue no. PAMM-015Z; SABiosciences-Qiagen) as per the manufacturer's

---

agonistic receptor DR4 using cell lysate from MS1 co-cultured either in the absence (control) or presence of differentiated trophoblast cells and cell lysate from co-cultured trophoblast cells (TC). **(F, G)** Densitometric analysis of the proteins from (F) using NIH ImageJ software after normalization with GAPDH. Data are representative of three independent biological replicates. **(H, J)** Western blot analysis of proteins from extrinsic (H) and intrinsic (J) apoptotic pathway proteins using lysates from MS1 cells co-cultured in the absence or presence of TC. **(G, I, K)** Densitometric analysis of the proteins from (G, I), respectively, using NIH ImageJ software after normalization with GAPDH. Data are representative of three independent biological replicates. Error bars represent SEM. **$P < 0.01$; ns, nonsignificant. Source data are available for this figure.

instructions. The PCR array included 84 SYBR Green-optimized primers related to endothelial cell function that were assessed in a 96-well format. RNA isolated from TS cells and day 6–differentiated trophoblast cells using TRIZOL reagent was further purified using an RNeasy Mini Kit (Qiagen). The concentration and quality of the RNA were measured using a NanoDrop 2000 Spectrophotometer (Thermo Fisher Scientific) followed by fractionation on a formaldehyde gel. The cDNA was synthesized using an $RT^2$ first strand kit (Qiagen) after genomic DNA elimination, and $RT^2$ SYBR Green qPCR Master Mix (Qiagen) was used for the real-time array. Normalization was done using the housekeeping gene, which showed no change in differentiated trophoblast cells compared with the TS cells. The fold change in gene expression was calculated by using online software provided by SABiosciences.

## Protein isolation and Western blotting

Total protein was extracted from cells using RIPA buffer (20 mM Tris–HCl, pH 7.5, containing 150 mM NaCl, 1 mM $Na_2$EDTA, 1 mM EGTA, 1% NP-40, 1% sodium deoxycholate, 2.5 mM sodium pyrophosphate, 1 mM $β$-glycerophosphate, 0.2 mM PMSF, and 1 mM sodium orthovanadate) supplemented with protease inhibitor mixture (Sigma-Aldrich). Protein concentration for each sample was estimated by using the Bio-Rad Protein Assay Reagent (Bio-Rad).

60–100 $μg$ of total proteins were fractionated by 10–12% SDS–PAGE (Bio-Rad) under reducing condition and were then transferred to PVDF membranes (MilliporeSigma). After blocking and incubation with primary and secondary antibody solution using standard protocol, an ECL reagent, Luminata Forte (MilliporeSigma), was used for chemiluminescence signal detection in a Biospectrum 810 imaging system (UVP, LLC). Densitometric analysis was done using NIH ImageJ software. Three biological replicates were used for each experiment. Antibodies used are detailed under "Antibodies" section.

## Antibodies

Anti-PECAM1 (2H8) and anti-endoglin (MJ7/18) were purchased from DSHB. Anti-CDH5 (AF1002) and anti-cytokeratin (C2562) were purchased from R&D Systems, and Sigma-Aldrich, respectively. These primary antibodies were used in 1:100 dilutions for flow cytometry and immunofluorescence. Anti-Qa2 (121711), PE-anti-HLAG (335905), APC-anti-CD105 (323207), and APC-anti-CD144 (348507) purchased from Biolegends, were used in 1:20 dilutions, and anti-HES1 (11988) antibody procured from Cell Signaling Technology was used in 1:50 dilutions for flow cytometry. For Western blotting, anti-PECAM1 (AF3628), anti-CDH5 (AF1002), anti-endoglin (AF1320), and anti-CRADD (AF4680) antibodies were purchased from R&D Systems, and were used in 1:1,000 dilutions. Primary antibodies obtained from Cell Signaling Technology were used in 1:1,000 dilutions for Western blotting and were as follows: anti-HES1 (11988), anti-KDR (9698), anti-c-KIT (3074), anti-integrin-$β$3 (13166), anti-caspase-8 (4790), anti-cleaved caspase-8 (8592), anti-caspase-3 (9665), anti-cleaved caspase-3 (9664), anti-BAX (2772), and anti-GAPDH (2118). Primary antibodies procured from Santa Cruz Biotechnology were used in 1:250 dilutions in Western blotting and were as follows: anti-COL18A1 (sc-16651), anti-BCL2 (sc-7382), anti-MMP9 (sc-6840), anti-PLAU (sc-14019), anti-BCL2 (sc-7382). Anti-DR4 (BML-SA225-0100)

antibody was purchased from Enzo Lifesciences, and was used in 1:1,000 dilutions for Western blotting.

For immunofluorescence, TRITC-conjugated rabbit anti-goat (T7028), rabbit anti-rat (T4280) secondary antibodies were purchased from Sigma-Aldrich. TRITC-conjugated goat anti-Armenian hamster (127-025-099) secondary antibodies were purchased from Jackson ImmunoResearch Laboratories. These antibodies were used in 1:2,000 dilutions. For flow cytometry, APC-conjugated donkey anti-goat (F0108), APC-conjugated goat anti-rat (F0113) secondary antibodies were purchased from R&D Systems. APC-conjugated goat anti-Armenian hamster (127-135-160), BV421-conjugated donkey anti-mouse (715-675-150), and BV421-conjugated goat anti-rabbit (111-675-144) secondary antibodies were purchased from Jackson ImmunoResearch Laboratories. These antibodies were used in 1:2,000 dilutions. For Western blotting, HRP-conjugated goat anti-rabbit (A120-101), goat anti-mouse (A90-116), and donkey anti-goat (A50-101) secondary antibodies were purchased from Bethyl Laboratories, and were used in 1:10,000 dilutions.

## Enzyme-linked immunosorbent assay

TS cells were cultured for 24 h after seeding in stemness condition, and differentiated trophoblast cells were cultured for 6 d after induction of differentiation in a 35-mm dish. Then for both types of cells, media were replaced with serum-free media and cultured for another 24 h. Serum-free conditioned medium was then collected and analyzed subsequently using mouse-specific ELISA kits. Expression of TRAIL (TNFSF10) and CX3CL1 was determined using a mouse (cat no. EMTNSF10; Thermo Fisher Scientific) and CX3CL1 quantikine ELISA kit (cat no. MCX310; R&D Systems), respectively, as per the manufacturers' instructions.

## ChIP assay

ChIP assay was performed with TS cells using a Simple ChIP Enzymatic Chromatin IP Kit (Cell Signaling Technology) as described previously (Saha & Ain, 2020). 10 $μg$ of chromatin DNA was incubated with either anti-HES1 or normal rabbit IgG or anti-RNA polymerase II antibody (Cell Signaling Technology) overnight at 4°C. This was followed by antibody–chromatin complex binding with protein G magnetic beads and further chromatin extraction as described previously (Saha & Ain, 2020). The extracted purified DNA was finally subjected to PCR amplification using primers specific for the HES1-binding sites (BS) on the promoter regions of murine *Cdh5*, *Pecam1*, and *Eng* DNA. A primer pair was used as negative control (NC) for each promoter/enhancer region at a location that does not contain any HES1-binding site. To confirm whether these amplified HES1-binding sites are transcriptionally active, similar ChIP analysis was also performed with anti-RNA polymerase II antibody (Cell Signaling Technology). Primers used for the ChIP assay are listed in Table S3.

## Co-culture of trophoblast and endothelial cells

TS cells were seeded on the cell culture inserts (cat no. 353102; BD Biosciences) at a density of 7.2 × $10^4$ cells/well, and differentiation

was induced with withdrawal of mitogens as described previously. Media were changed every 24 h from day 2 to day 5 post differentiation induction. On day 5 of differentiation, these cells were co-cultured with MS1 endothelial cells seeded on the companion plates (cat no. 353502; BD Biosciences) at a density of $3 \times 10^5$ cells/well 24 h before co-culturing. Cells were co-cultured for 48 h at 37°C in a humidified incubator with 5% $CO_2$. For control experiment, MS1 cells were grown in the companion plate, keeping all conditions identical but without the inserts and in media containing TS basal media (without mitogens) and MS1 media in 1:1 ratio. After 48 and 72 h, co-cultured cells were processed either for both annexin V–PI staining using early apoptosis detection kit (cat no. 6592; Cell Signaling Technology) by flow cytometry or for protein isolation and Western blotting for detection of apoptosis markers.

### Early apoptosis detection assay using annexin V–PI staining and flow cytometric analysis

MS1 cells from the companion plates of the co-culture experiment were trypsinized and harvested at a concentration of $10^6$ cells/250 μl. Cells were then stained with annexin V and PI (cat no. 6592; Cell Signaling Technology) as per manufacturer's instructions. Percentage of cells undergoing early apoptosis was analyzed using a flow cytometer (LSR Fortessa; BD).

To analyze TRAIL-induced apoptosis, both endothelial cells (MS1) and differentiated trophoblast cells on day 5 of differentiation were treated with recombinant murine TRAIL (Peprotech) at a concentration of 6, 12, and 25 ng for 24 h at 37°C humidified incubator followed by annexin V and PI staining. Percentage of cells undergoing early apoptosis was analyzed using a flow cytometer (LSR Fortessa; BD).

### Statistical analysis

Data are represented as mean ± SE or SEM. A two-tailed, unpaired Student $t$ test was used to compare between groups. $P < 0.5$ is considered to be statistically significant for all experiments. Statistical evaluations were done using GraphPad Prism (version 6.0) software.

## Supplementary Information

## Acknowledgements

We would like to thank Prof. Janet Rossant, Hospital for Sick Children, Toronto, Canada, for providing the TS cells. Authors thank flow cytometry core facility of CSIR-IICB for aiding in data acquisition. This work was supported by seed grant from Council of Scientific and Industrial Research-Indian Institute of Chemical Biology. Financial support from Department of Biotechnology-Research Associateship programme in Biotechnology and Life Sciences to Dr. M Paul is gratefully acknowledged.

## Author Contributions

M Paul: conceptualization, formal analysis, investigation, methodology, and writing—original draft.
S Chakraborty: conceptualization, formal analysis, investigation, and methodology.
S Islam: investigation.
R Ain: conceptualization, resources, data curation, formal analysis, supervision, funding acquisition, investigation, and writing—review and editing.

## Conflict of Interest Statement

The authors declare that they have no conflict of interest.

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
