## [Reviewer comments · Life Science Alliance]

Life Science Alliance

Trans-differentiation of trophoblast stem cells: implications in placental biology

Madhurima Paul, Shreeta Chakraborty, Safirul Islam, Rupasri Ain

DOI: <https://doi.org/10.26508/lsa.202201583>

Corresponding author(s): *Dr. Rupasri Ain (CSIR-Indian Institute of Chemical Biology)*

Review Timeline:

Submission Date:	2022-06-29
Editorial Decision:	2022-08-08
Revision Received:	2022-11-08
Editorial Decision:	2022-12-01
Revision Received:	2022-12-12
Accepted:	2022-12-13

Transaction Report:

August 8, 2022

Re: Life Science Alliance manuscript #LSA-2022-01583

Dr. Rupasri Ain
CSIR-Indian Institute of Chemical Biology
Division of Cell Biology and Physiology
4, Raja S.C Mullick Road
Jadavpur
Kolkata, West Bengal 700032
India

Dear Dr. Ain,

Thank you for submitting your manuscript entitled "Trans-differentiation of trophoblast stem cells: implications in placental biology" to Life Science Alliance. The manuscript was assessed by expert reviewers, whose comments are appended to this letter. We invite you to submit a revised manuscript addressing the Reviewer comments.

Thank you for this interesting contribution to Life Science Alliance. We are looking forward to receiving your revised manuscript.

Sincerely,

B. MANUSCRIPT ORGANIZATION AND FORMATTING:

Reviewer #1 (Comments to the Authors (Required)):

The manuscript provides evidence about an interesting aspect about the trophoblast biology. The evidence provided in this manuscript indicates a trans differentiation ability of mouse TSCs to acquire endothelial cell like phenotype. This is important considering that Trophoblast cells and endothelial cells are in proximity at the maternal-fetal interface and a subset of trophoblast cells (end-vascular) can remodel uterine artery to ensure proper pregnancy progression. The manuscript is well written. I have following comments:

1. Authors should show whether the trans differentiation ability of TSCs are increased upon factors that promote or support endothelial niche. For example, medium contains growth factors like VEGF, bFGF etc. This experiment should also be done with HES1-depleted TSCs.
2. What is the HES1 expression pattern in TSCs, which does not transdifferentiate to endothelial phenotype? Authors should come up with a better explanation with experimental evidence.
3. The MS1 endothelial cells (established from pancreas) does not properly represent the endothelial cells at the maternal-fetal interface. For example, trophoblast cells are in close proximity at the fetal vasculature at the labyrinth zone of a mouse placenta. Is there an apoptotic effect on labyrinth endothelial cells? Authors should isolate endothelial cells from mouse placenta and uterus and perform the co-culture experiment with endothelial cells from both compartments to show if there is a differential apoptotic response.

Reviewer #2 (Comments to the Authors (Required)):

In this study by Paul et al, the authors investigate the endothelial cell genesis during mouse trophoblast stem cell differentiation. Authors indicate that trophoblast-vasculogenic mimicry involves acquisition of trophendothelial phenotype during mouse TS cell differentiation and related with trophoblast-induced apoptosis of endothelial cells. The manuscript is well written, interesting and easy to follow. The experimental design, methodology, statistical analysis, and result presentation are convincing. However, there are concerns about the manuscript that have been detailed below:

Authors should describe the transition of trophendothelial phenotype in human TS cells.

Does uterine contribution is important for trophendothelial phenotype?

Typo in page 9 nm should be nM.

Please assess the expression of endothelial marker Kdr in the absence of Hes1.

Authors can include some recent reports related to endovascular trophoblast in the introduction section: PMID: 34819240 and PMID: 35328368.

Reviewer #3 (Comments to the Authors (Required)):

This is an interesting manuscript with a potential expand our understanding of TS cells differentiation, where there are significant gaps in our understanding of the differentiation process. The investigators have conducted a variety of complementary experiments to support their conclusion about the expression of a subset of genes involved in the differentiation of TS cells into invasive trophoblast cells. However, the manuscript has several gaps and conclusions are not well supported by the data. Appropriate controls are missing from some of the experiments. Some of the obvious questions are not addressed to strengthen the manuscript. Additionally, the data presented and conclusions are not particularly innovative and parts of the information, such as the role of HES1 in the differentiation of TS cells and the down regulation of apoptotic pathways in differentiated cells is already reported in the literature. In order for the manuscript to be more convincing to researchers familiar with the field they need to address multiple issues some of which are listed below.

Unfortunately, I cannot recommend this manuscript for publication in its current form.

Specific comments:

Cytokeratin (CK-BV42) may not be the best marker to correctly identify TS cells. Investigators need to use additional markers to convince the reader that they are using good quality TS cells for their experiments.

Immuno-stained TS cells appear dispersed which is contrary to the tightly packed colonies that is a hallmark of TS cells (Fig. 1). This is likely due to the fact that the researchers are looking at TS cells 24 hours after plating, while TG cells remain in culture for 6 days. A better experiment will be to keep TS cells and differentiated cells in culture for the same number of days and then analyze them for expression of various genes.

Transcriptomic analysis is conducted on a mixed population of 6 days differentiated TS cells. TS cells differentiate into number of different cell types. As shown in figure 1, only a small subset of the differentiated cells showed expression of endothelial markers. Thus, in order for the transcriptomic data to be more meaningful, sorting of the differentiated cells is needed to select cells that express endothelial markers of interest and then look at the larger array of endothelial specific markers in that sub-population of cells.

TS cells differentiation led to about 85% reduction in HES1 protein level (Fig 5), however the percentage of cells that showed expression of endothelial markers was about one third (as shown in figure 1). Does this mean, reduction in HES1 levels results in the differentiation of TS cells to other types of differentiated cells as well? Time course studies are needed to better correlate down regulation of HES1 with the expression of endothelial specific markers the authors are focusing on. Western blotting showing just one time point is just not convincing enough.

Co-culture of differentiated TS cells with endothelial cells needs a better control. Using basal TS medium without the cells is not a convincing control. Sorting differentiated cells into populations that express endothelial markers and ones that don't express such markers and then using them separately for the co-culture experiments will be more convincing.

A relatively small percentage of the co-cultured endothelial cells undergo apoptosis after two days of treatment. Time course assays of co-culturing experiments are needed to get a better understanding of the effect of differentiated cells on co-cultured endothelial cells.

Differentiated TS cells have different ploidy levels. Ploidy level correlates to their function. Addressing ploidy levels of the cells that express their markers of interest will expand the appeal of the manuscript.

Additional minor comments:

Incomplete sentence

In line with this, precocious down-regulation of HES1 protein during TS cell differentiation led to significant ($p < 0.01$) up-regulation of CDH5 (1.3 fold) and ENG (2.24 fold) protein levels, upon (Fig. 5F and 5G).

Confusing logic

Pages 5 and 6.

Expression of the endothelial markers were observed exclusively in differentiated trophoblast cells and in the TS cells (Fig. 1A-1C).

TS cells showed almost negligible population (0.1%-0.2%) expressing both the trophoblast and endothelial markers while majority of the population expressed only Ck.

Authors thank all reviewers for very useful and constructive criticism. We have taken note of each concern raised by the reviewers and have performed relevant experiments as per their advice and incorporated the revised data in the manuscript as described below.

The new incorporations as well as modified sentences have been marked red in the text of the manuscript.

- Figure 1 has been modified. Images in A, B and C have been replaced with new images showing tightly packed colonies of TSCs as per advice of reviewer 3.
- In Figure 7, E has been incorporated showing apoptotic endothelial cell population co-cultured with TC cells for 72h, as per advice of reviewer 3. Accordingly, figure 7 E, F, G, H, I and J have been changed to F, G, H, I, J and K.
- Figure S1 has been changed to figure S5 and figure S2 to S7.
- Figure S1, S2, S3, S4 and S6 have been newly added.
- New figure legends corresponding to the new additions have been incorporated.
- Four new references, # 17, # 22, # 23 and # 43 have been added and the reference numbers have been changed accordingly.

Reviewer 1:

1) The reviewer raised question whether the trans-differentiation ability of TSCs are increased upon factors that promote or support endothelial niche, for example, medium containing growth factors like VEGF, bFGF etc. and also that this experiment should also be done with HES1- depleted TSCs.

As per reviewer's suggestion, change in trans-differentiation ability of TSCs was checked in presence of endothelial growth promoting factors VEGF₁₆₅ and bFGF. Results from these experiments have been shown in supplementary **Figure S2 (New)**. We have also included results from similar experiment done with HES1- depleted TSCs in supplementary **Figure S6**. New text has been inserted accordingly in the manuscript.

Experimental procedure has been included in the "Methods" section under "Cell Culture and differentiation" and under "RNA interference" marked in red.

2) The reviewer asked to come up with experimental evidence about the HES1 expression pattern in TSCs, which does not transdifferentiate to endothelial phenotype.

As per reviewer's suggestion, flow-cytometric analysis has been done to assess cell population expressing HES1 and/or endothelial marker CDH5 in both TS and differentiated trophoblast cells. Results from these experiments have been shown in supplementary **Figure S4**. New text has been inserted accordingly in the manuscript.

3) The reviewer has shown concern that the MS1 endothelial cells does not represent the endothelial cells at the maternal-fetal interface. The reviewer has suggested isolation of endothelial cells from mouse placenta and uterus and perform the co-culture experiment with endothelial cells from both compartments to show if there is a differential apoptotic response.

MS1 cells are murine microvascular endothelial cells possessing all the salient endothelial phenotypes like in vitro capillary formation (doi: 10.1074/jbc.M116.742627), ability to form endothelial tumors like angiosarcomas, hemangiomas ([https://doi.org/10.1016/S0002-9440\(10\)65015-8](https://doi.org/10.1016/S0002-9440(10)65015-8)), known to express Vascular endothelial growth factor or VEGF receptor KDR ([https://doi.org/10.1016/S0002-9440\(10\)65015-8](https://doi.org/10.1016/S0002-9440(10)65015-8)). MS1 cells have been used as a model to study endothelial function in several reports [(doi: 10.1074/jbc.M116.742627; doi.org/10.1016/j.ajpath.2018.01.015; 10.1152/ajplung.90477.2008, ANTICANCER RESEARCH 25: 1851-1864 (2005)]. Besides, MS1 cells share similarities with primary human umbilical vein endothelial cells (HUVEC) in terms of vascular lumen formation via their basal cell surfaces (10.1371/journal.pone.0004132). Hence, we have chosen MS1 cells to study effect of invasive trophoblast cells in vitro.

Reviewer 2

1) The reviewer has asked to describe the transition of trophendothelial phenotype in human TS cells.

Human TS cells established by a research group in Japan on 2018. We have tried acquiring them several times writing to the PI but it was not shared with our group. These cells are not commercially available. So, we used human JEG3 cell line and demonstrated

acquisition of trophendothelial phenotype by this cell line upon induction of mesenchymal type. We have addressed the concern of the reviewer using flow cytometric analysis of cell population co-expressing human endothelial (CD144, CD105) and trophoblast (HLA-G) markers in human trophoblast JEG3 cells. Results have been included in supplementary **Figure S1 (new)**. Text pertaining to this experiment has been inserted accordingly in the manuscript. Experimental procedure has been included in the “Methods” section under “Cell Culture and differentiation” marked in red.

2) The reviewer has asked whether uterine contribution is important for trophendothelial phenotype.

This is an interesting question. However, this is beyond the purview of this manuscript.

3) The reviewer has asked to change the typing error in page 9 from nm to nM.

This typing error has been rectified in the manuscript.

4) The reviewer has enquired about the expression of endothelial marker Kdr in the absence of HES1.

We performed western blotting using lysates from trophoblast cells transfected with either scrambled or Hes1 siRNA. But, KDR expression was not inversely regulated by HES1 as shown in the figure below.

5) The reviewer has asked to include some recent reports related to endovascular trophoblast in the introduction section: PMID: 3419240 and PMID: 35328368.

These references have been incorporated in the introduction section.

Reviewer 3

1) The reviewer has asked the investigators to use additional markers other than Cytokeratin (Ck) to convince the reader that they are using good quality TS cells for their experiments.

As pointed out by the reviewer, we have analyzed population of trophoblast cells expressing another trophoblast marker, Qa2 and compared it with Ck expressing population using flow-cytometry. Murine Qa2 is functionally similar to the human trophoblast marker HLA-G and has been reported to be expressed by preimplantation embryos (Dietz, S., Schwarz, J., Velic, A., González-Menéndez, I., et.al (2021)). Using flow cytometric analysis we demonstrated that the percentage of trophoblast cells express Qa2 or Ck, in stemness and differentiated conditions are similar, as shown below. Hence, we have used Ck as trophoblast marker for this study.

1a) The reviewer suggested to keep TS cells and differentiated cells in culture for same number of days and then analyze them for expression of various genes.

As per suggestion of the reviewer we have cultured both the TS and TC cells for 6 days and have repeated the immunostaining of the cells for the endothelial markers. Figure1 A, B and C have been replaced with new images showing TS cells in tightly packed colonies.

2) The reviewer indicated that as in figure 1, only a small subset of the differentiated cells showed expression of endothelial markers, thus in order for the transcriptomic data to be more meaningful, sorting of the differentiated cells is needed to select cells that express endothelial markers of interest and then look at the larger array of endothelial specific markers in that sub-population of cells.

Our transcriptomic analysis was done with the entire population of trophoblast stem cells and differentiated trophoblast cells, to check trophoblast cell differentiation-associated changes in expression of markers other than exclusive trophoblast markers and in particular endothelial markers. Our question was not to find out “trophendothelial” gene expression hence we didn’t think of sorting the differentiated cells to select only the “trophendothelial” population.

3) The reviewer indicated that as in figure 1, the percentage of cells that showed expression of endothelial markers was about one third, while TS cell differentiation led to about 85% reduction in HES1 protein level (Fig5). The reviewer has suggested to do time course studies to better correlate down regulation of HES1 with the expression of endothelial markers the authors are focusing on.

As per reviewer’s suggestion we have performed time course study showing expression pattern of HES1 and two of the representative endothelial markers CDH5 and PECAM1. Results have been incorporated in supplementary **Figure S3** and new text has been inserted accordingly in the manuscript.

4) The reviewer has asked for a better control in the experiment showing co-culture of differentiated TS cells with endothelial cells as using basal TS medium without the cells didn’t seem a convincing control. Reviewer has suggested to sort differentiated cells into

populations that express endothelial markers and the ones that don't express such markers and then to use them separately for the co-culture experiments.

The main objective of our co-culture experiment was to check the potential of differentiated trophoblast cells to induce apoptotic death in the endothelial cells in their close vicinity and not to show that specifically the "trophendothelial" population was mediating the same. Hence, we have kept endothelial cells without the differentiated cells as our co-culture control.

5) The reviewer has asked to show time course assays of co-culturing experiments to get a better understanding of the effect of differentiated cells on co-cultured endothelial cells as a relatively small percentage of the co-cultured endothelial cells undergo apoptosis after 2 days of treatment.

As has been suggested by the reviewer, we have extended our co-culture experiment till 72h and have included the results as **Figure 7E**. New text has been inserted accordingly in the manuscript.

6) The reviewer has asked to address ploidy levels of the cells that express their markers of interest that will expand the appeal of the manuscript.

The main objective of our study was to find out differentiation associated acquisition of endothelial phenotype and we didn't focus on the type of differentiated cells that are acquiring the same. So, it is beyond the scope of our present study to address ploidy levels of "trophendothelial cells".

Additional minor comments:

1) The reviewer has pointed out incomplete sentence: ' In line with this, precocious down-regulation of HES1 protein during TS cell differentiation led to significant ($p < 0.01$) up-regulation of CDH5 (1.3 fold) and ENG (2.24 fold) protein levels, upon (Fig.5F and 5G)'.
The sentence has been corrected and marked in red.

2) The reviewer has pointed out a confusing logic in pages 5 and 6 as: a) 'Expression of the endothelial markers were observed exclusively in differentiated trophoblast cells and in the TS cells (Fig.1A-1C)', b) 'TS cells showed almost negligible population (0.1%-0.2%)

expressing both the trophoblast and endothelial markers while majority of the population expressed only Ck'.

It was a typing error and has been corrected.

December 1, 2022

RE: Life Science Alliance Manuscript #LSA-2022-01583R

Dr. Rupasri Ain
CSIR-Indian Institute of Chemical Biology
Division of Cell Biology and Physiology
4, Raja S.C. Mullick Road
Jadavpur
Kolkata, West Bengal 700032
India

Dear Dr. Ain,

Thank you for submitting your revised manuscript entitled "Trans-differentiation of trophoblast stem cells: implications in placental biology". We would be happy to publish your paper in Life Science Alliance pending final revisions necessary to meet our formatting guidelines.

- please add a callout for Figure 4H and Figure 7K to the main manuscript text
- in your figure legend, you label your Source Data as Figure S7-please remove this from the figure legend; it is sufficient to have your source data uploaded as a separate file, and we do not need it designated in the figure legend - please upload one source data file per figure, labeled "Source Data for Figure X"
- Figure 8 can instead be uploaded as a Graphical Abstract, if you prefer

A. FINAL FILES:

B. MANUSCRIPT ORGANIZATION AND FORMATTING:

Sincerely,

Reviewer #1 (Comments to the Authors (Required)):

The revised manuscript addresses most of my initial concerns. I do not have any more comments.

Reviewer #2 (Comments to the Authors (Required)):

The authors have satisfactorily addressed most of my concerns. I consider the manuscript acceptable for publication.

December 13, 2022

RE: Life Science Alliance Manuscript #LSA-2022-01583RR

Dr. Rupasri Ain
CSIR-Indian Institute of Chemical Biology
Division of Cell Biology and Physiology
4, Raja S.C> Mullick Road
Jadavpur
Kolkata, West Bengal 700032
India

Dear Dr. Ain,

Thank you for submitting your Research Article entitled "Trans-differentiation of trophoblast stem cells: implications in placental biology". It is a pleasure to let you know that your manuscript is now accepted for publication in Life Science Alliance. Congratulations on this interesting work.

DISTRIBUTION OF MATERIALS:

Again, congratulations on a very nice paper. I hope you found the review process to be constructive and are pleased with how the manuscript was handled editorially. We look forward to future exciting submissions from your lab.

Sincerely,
